# Exploring solid-phase proximity ligation assay for survivin detection in urine

**Jan Gleichenhagen** [1]*, **Christian Arndt**[2], **Swaantje Casjens**[1], **Carmen Töpfer**[1], **Holger Gerullis**[3], **Irina Raiko**[1], **Dirk Taeger**[1], **Thorsten Ecke**[4], **Thomas Brüning**[1], **Georg Johnen**[1]

**1** Institute for Prevention and Occupational Medicine of the German Social Accident Insurance, Institute of the Ruhr-University Bochum (IPA), Bochum, Germany, **2** Department of Urology, HELIOS Hospital, Krefeld, Germany, **3** University Hospital for Urology, Klinikum Oldenburg, Oldenburg, Germany, **4** Department of Urology, HELIOS Hospital, Bad Saarow, Germany

* Jan.Gleichenhagen@ipa.ruhr-uni-bochum.de

**Data Availability Statement:** All relevant data are within the manuscript and its Supporting Information files.

**Funding:** The authors received no specific funding for this work.

## Abstract

Urine-based biomarkers are a rational and promising approach for the detection of bladder cancer due to the proximity of urine to the location of the tumor site and the non-invasive nature of its sampling. A well-known and highly investigated biomarker for bladder cancer is survivin. For detection of very small amounts of urinary survivin protein a highly sensitive assay was developed. The assay is based on the immuno-PCR technology, more precisely a solid-phase proximity ligation assay (spPLA). The limit of detection for the survivin spPLA was 1.45 pg/mL, resulting in an improvement of the limit of detection by a factor of approximately 23 compared to the previously in-house developed survivin ELISA. A key step in development was the initial isolation of survivin by a molecular fishing rod based on magnetic beads. Interfering matrix compounds pose a special challenge for further analytical application, but can be overcome by this isolation step. The assay is designed to work with only 500 μL of voided urine. The survivin spPLA showed a sensitivity of 30% and specificity of 89% for bladder cancer detection in this study of 110 bladder cancer cases and 133 clinical controls. Moreover, the results demonstrated again that survivin is a useful complementary marker in combination with UBC® Rapid by increasing the overall sensitivity to 70% with a specificity of 86%. Although the performance for detection of bladder cancer was rather low, the herein developed assay might serve as a new tool for survivin biomarker research in diverse human fluids, even if the biological matrix is complex or survivin is only present in small amounts.

## Introduction

Bladder cancer ranks 10th among the most common cancers in the world [1,2]. Especially in Europe and other developed countries, the incidence of bladder cancer is still rising. Improved prevention, early diagnosis, and treatment helped to reduce mortality worldwide [1,2].

The bladder is a hollow muscular organ collecting and storing urine originated from kidneys until micturition. Urothelial cells are specialized transitional epithelial cells of the bladder

**Competing interests:** The authors have declared that no competing interests exist.

and urinary tract capable of delimiting urine by forming an impenetrable barrier. Those cells are constantly exposed to environmental, potentially mutagenic agents that are filtered into the urine by the kidneys [1].

Typically, bladder cancer is diagnosed by cystoscopy followed by pathological examination of suspicious tissue. It is a time-consuming and expertise-based invasive method, which can be painful for the patients. Non-invasive diagnostic methods can improve and simplify diagnosis and simultaneously prevent observer-biased results. Being minimally or non-invasive is a key characteristic of biomarkers measured in easily accessible body fluids like blood, plasma, urine or saliva [3]. Because of its close proximity to the target organ, it is of general acceptance that urine can serve as a good source for bladder cancer-specific biomarkers [4–8].

Examples of biomarkers for bladder cancer detection with potential for translation into clinical routine or already in use are: UBC® Rapid, NMP22®, UroVysion, UroSEEK, Xpert® Bladder Cancer Detection, Assure MDx, Cxbladder, URO17™ and CellDetect®. Molecular targets of those assays comprise abnormal protein levels, aneuploidy, mutation signatures, unusual mRNA expression or metabolic signature [7–11]. The well-known ImmunoCyt/uCyt + assay is no longer commercially available.

Survivin is an evolutionarily conserved protein that is essential for cell division and able to inhibit cell death. In adults, it is absent or rarely present in healthy tissue, but it is specifically expressed in tumor tissue. Survivin serves a variety of functions, mainly in association with other proteins, and is acting as an adaptor protein transporting the interaction partners to their destination. [12]. It is directly or indirectly involved in numerous pathways required for tumor maintenance and moreover plays also an important role in cell cycle, apoptosis, mitosis, proliferation, angiogenesis, and consequently cancer formation and progression [13,14]. Hence, survivin is of high interest to cancer-related biomarker research and targeted cancer therapy [12,15].

Different approaches have been investigated to make use of survivin as a biomarker in human fluids for cancer detection. In most cases researchers focused on survivin protein and/or survivin mRNA (BIRC5). Independently of the molecular target, both approaches showed varying sensitivities and specificities among different studies [16–21]. On the other hand, the presence of survivin in bladder cancer tissue has been consistently documented in several publications [22,23]. Current studies explore survivin as a molecular target for therapy, thereby elucidating whether the subcellular location of survivin may be associated with tumor aggressiveness [24,25].

Previously, a new survivin ELISA based on a polyclonal antibody was successfully developed in-house [19]. The limit of detection seemed to be the bottleneck of the ELISA resulting in an unsatisfactory sensitivity for bladder cancer detection. To overcome this limitation, the approach for survivin detection was fundamentally changed. To the best of our knowledge, this is the first survivin assay using an immuno-PCR approach that is based on the solid-phase proximity ligation assay (spPLA) technique.

Originally, immuno-PCR was introduced 1992 by Sano et al. [26]. Because of technical challenges this method remained a niche technology in the field of protein quantification for a long time. Only over the last decade this technique appeared to become more applicable in protein science. Possible reasons might be that DNA-antibody coupling, modified oligonucleotide labeling, and detailed protocols became accessible to a broader range of researchers [27–31]. Nowadays, the basic technique of immuno-PCR evolved into different sub-techniques adapted to their intended use. Common applications comprise protein-protein interaction, protein localization and also protein quantification in biological samples like serum, plasma, cerebrospinal fluid, cell culture media, and lysates of cells or tissues [30,32]. All applications share the central feature of transferring a specific protein signal to an amplifiable DNA signal.

The proximity ligation assay (PLA) is a method suitable for protein quantification and localization. It can be further distinguished into a homogenous and a solid-phase based approach [32,33]. The solid-phase proximity ligation assay (spPLA) is a particularly suitable approach if the target needs to be enriched and/or isolated from a complex matrix like urine [6,34]. The principle of the spPLA is based on an initial target separation. The target protein is trapped from solution by binding to an antibody attached to a solid phase, e.g., magnetic beads. The target detection is facilitated by a pair of proximity probes. Proximity probes are composed of a unit, usually an antibody, able to specifically bind the target molecule and an attached single-stranded DNA moiety. In case of simultaneous target binding by these proximity probes, DNA strands came into close proximity and are joined by ligation with the help of a DNA connector molecule. A newly chimeric DNA strand is formed and can serve as a surrogate marker for specific detection of the target by an amplification method like real-time PCR. Thus, a singular protein signal gets transformed to an amplifiable DNA signal [32,34]. Only few attempts have been investigated to explore the use of spPLA for protein detection in urine.

Therefore, the use of the spPLA technique for survivin detection in urinary samples from patients with bladder cancer was explored. It was hypothesized that the feature of transforming the protein signal into an amplifiable signal could overcome the lack of sensitivity associated with the classical ELISA approach and thereby facilitate the use of survivin as a biomarker for bladder cancer detection.

## Materials and methods

### Study population

Between January 2014 and July 2015, a total of 290 patients were recruited at the Lukaskrankenhaus (Neuss, Germany). Because of a prior bladder cancer diagnosis, 46 patients were excluded. The remaining 244 participants included 111 patients suffering from bladder cancer (cases) and 133 control patients visiting for other reasons than bladder cancer or urologic disease (clinical controls). The initial diagnosis of bladder cancer was based on cystoscopy and confirmed by histological and immunohistochemical examination of resected tissue. Tumors have been assigned as low- or high-grade tumors according to the 2004 WHO classification [58]. Urine was collected before transurethral resection. Cases and clinical controls were matched for age and sex. The study was approved by the ethics committee of the Landesärztekammer Brandenburg No. AS 147(bB)/2013 (17 November 2013). All participants of the study provided written informed consent. The methods were carried out in accordance with the relevant guidelines and regulations.

### Urine sample collection

Midstream urine samples were collected at the urology department of the Lukaskrankenhaus Neuss, Germany. Urinary samples were processed immediately or stored at +4°C for a maximum of 4 h. Urine status was assayed by routine dipstick analysis. For the UBC® Rapid assay, three drops of fresh urine were used prior to further processing of a urine sample. A 2 mL aliquot urine were taken and 1 mL of this aliquot was reserved for the spPLA. Samples were stored at −20°C until further analyses were carried out. For analysis, the samples were shipped on dry ice to the IPA in Bochum.

### UBC® Rapid assay

The UBC® Rapid assay (concile GmbH, Freiburg/Breisgau, Germany) was performed according to the manufacturers' instructions as previously described [35]. The test cartridges were

read out by the photometric point-of-care (POC) system concile® Ω100 reader (concile GmbH) according to the manufacturers' instructions, allowing a quantitative analysis of the test results.

## Coupling of survivin protein to NHS-column

For isolation of antigen-purified anti-survivin antibody sera of immunized rabbits were used and purified His10-survivin as described earlier [19]. First, we covalently linked the purified survivin protein to an NHS-column (GE Healthcare Bio-Sciences AB, Uppsala, Sweden) according to manufacturer's instructions. Briefly, His10-survivin was desalted against coupling buffer (200 mM NaHCO3, 500 mM NaCl, pH 8.3) via PD-10 column (GE Healthcare Bio-Sciences). The NHS-column was initially washed with 1 mL of ice-cold HCl and immediately following 1 mL binding solution containing 1 mg/mL His10-survivin in coupling buffer. After incubation at +4˚C for 4 h the column was alternatingly washed and blocked according to the manufacturer's instructions. Coupling efficiency was determined by absorption at 280 nm after desalting the flow through via PD-10 column. For equilibration the survivin column was washed with 10 column volumes of PBS.

## Isolation and preparation of antigen-purified survivin antibody

For antibody purification 30 mL of serum was loaded onto the equilibrated survivin column linked to a Next Generation Chromatography System (Bio-Rad). The column was washed with PBS and the purified antibodies were eluted with 100 mM glycine (pH 2.8). Purified anti-survivin antibodies were immediately dialyzed against PBS yielding a final IgG concentration of 0.6 mg/mL in a total volume of 5 mL as determined by absorption at 280 nm. The purified survivin antibody was biotinylated with 33× molar excess of NHS-Biotin (Thermo Fisher, Waltham, MA, USA) in 10 mM NaHCO3 (pH 8.4), incubated for 3 h at room temperature and finally dialyzed against PBS.

## Preparation of proximity probes

The proximity probes were prepared according to manufacturer's instructions including guidelines for selecting biotinylated antibodies. Briefly, the antigen-purified polyclonal antibody against survivin was diluted to 200 nM (30 μg/mL). Equal molar amounts of 3'Prox-Oligo or 5'Prox-Olgio were combined with a separate portion of diluted survivin antibody and gently mixed. After incubation for 60 minutes at room temperature, the assay probe storage buffer was added followed by an incubation for 20 minutes at room temperature. Finally, the assay probes were stored at –20˚C for up to 6 months.

## Preparation of antibody-functionalized magnetic beads

The antibody-functionalized magnetic beads representing the solid phase for the survivin spPLA were generated according to the detailed protocol provided by Nong and colleagues [28]. In brief, magnetic beads (Dynabeads® MyOne™ Streptavidin T1, Life Technologies, Carlsbad, CA 92008 USA) were resuspended in the stock tube prior transfer of 100 μL of beads (10 mg/mL) to a new 1.5 mL reaction tube. First, storage solution was removed by magnetic separation followed by two washing steps with PBST (PBS containing 0.05% (vol/vol) Tween-20). The antigen-purified survivin antibody was diluted to 50 nM in 200 μL of storage buffer (PBS with 0.1 mg/mL BSA) and mixed with the prepared beads. The reaction was incubated on an end-over-end rotator at a speed of 16–20 rpm for one hour at room temperature. Afterwards, the magnetic beads were isolated using a magnetic stand. The supernatant was

discarded and the beads were washed three times with PBST. Finally, the functionalized beads were resuspended in 200 μL of storage buffer and stored at +4˚C.

## Initial spike-in experiments

Spike-in experiments were performed to pre-test the method's ability to detect survivin in the intended biomatrix. For initial experiments with the TaqMan Protein Assay® Kit, which performs a homogenous proximity ligation assay, we tested urine sample volumes of 2–12 μL. The assay was performed and analyzed according to the manufacturer's protocol. Initial experiments with the TaqMan Protein Assay® Kit performing a solid-phase proximity ligation assay used 200 μL of urine sample in a total volume of 2 mL PBS buffer for neutralization. All subsequent steps—binding to the magnetic beads, washing, binding of the proximity probes, ligation, and real-time PCR—were performed as described below in Sections 4.9 and 4.10.

## Preparation of urinary samples

For measuring survivin in urinary samples, 500 μL voided urine was concentrated by using Vivaspin 500 spin columns (Sartorius, Stonehouse, Gloucestershire, UK) to 50 μL. The remaining sample were used to resuspend any residual urinary amount of soluble protein within the Vivaspin 500 spin column. The sample was divided into two 20 μL aliquots and transferred to separate reaction wells of a MicroAmp® Optical 8-Tube Strip (Applied Biosystems, Life Technologies) for measurement in duplicate. All following steps are performed within the same reaction well avoiding analyte loss by sample transfer. To neutralize the sample for appropriate binding conditions 175 µl PBS containing 0.05 mg/mL BSA were added. Next, 1 μL of anti-survivin antibody-functionalized magnetic beads were resuspended in 5 μL of PBS containing 0.05 mg/mL BSA before added to the binding reaction. Finally, the anti-survivin antibody functionalized magnetic beads were allowed to bind free survivin within the reaction mixture on an end-over-end rotator at a speed of 16–20 rpm at +4˚C for 16 h. The reference curve was generated by using different concentrations of recombinant survivin in PBS containing 0.05 mg/mL BSA in equivalent amounts like the urinary samples. The reference samples were also measured in duplicate.

## Solid-phase proximity ligation assay

The binding reaction procedure was mainly adopted from the TaqMan Protein Assay® kit manufacturer's specifications with minor modifications and is described in more detail below. After incubating of urinary samples with antibody-functionalized magnetic beads, the beads and consequently the bound survivin was isolated by magnetic separation and washed three times with 250 μL PBST. Next, the magnetic beads where mixed with 10 μL detection buffer (PBS-BSA 0.05%) containing 0.1 μL of each proximity probe A and B. The reaction mixture was incubated for 3 h at room temperature on an end-over-end rotator at a speed of 15 rpm. After incubation, the beads were washed three times with 80 μL PBST and resuspended in 90 μL ligation mix containing ligation buffer, the connector, and the ligase according to manufacturer's instructions. The connector overlaps both terminal DNA strands and could thereby simultaneously bind both probes. Subsequently, the ligase reaction covalently joined the bridged DNA-strands. The ligation reaction was performed in a thermocycler for 10 minutes at +37˚C following 10 minutes at +10˚C. Finally, the beads were separated and resuspended in 20 μL qPCR-mix of the TaqMan Protein Assay® kit. For quantitative real-time PCR (qPCR) a 7900 HT Fast Real-Time PCR System (Thermo Fisher Scientific) was used according to the manufacturer's instructions. A 4-parameter curve fitting was performed with Prism 8

(GraphPad Software, Inc., San Diego, CA, USA) to provide a reference curve for interpolation of unknown survivin concentrations in urine samples.

## Statistics

Statistical analyses were performed with SAS/STAT and SAS/IML software version 9.4 (SAS Institute Inc., Cary, NC, USA) or Prism 8. Plots were generated with Prism 8. Median and inter-quartile ranges were used to describe the distribution of continuous variables. Groups were compared using the non-parametric Wilcoxon signed-rank test. The performance of survivin spPLA was evaluated by receiver operating characteristic (ROC) analysis. Logistic regression was performed to assess the risk of survivin concentrations being above the cutoff.

## Results

### Preparation of the survivin antibody

The basic prerequisite for developing a solid-phase proximity ligation assay is an epitope recognition by some kind of a DNA-tagged binder (i.e., antibody) followed by amplification of this DNA-tag. Accordingly, the specificity of this binder is critical in order to avoid false-positive signals. Therefore, an antigen-purified antibody is required for application in spPLA. For antigen-based purification of the antibody, affinity-purified his-tagged survivin was coupled to an NHS-column following manufacturer's instructions. The coupling efficiency was 89.2%, thereby meeting basic requirements for an antigen-based purification of survivin from rabbit sera. A total volume of 5 mL with a concentration of 0.6 mg/mL specific survivin antibody could be isolated utilizing standard antibody protocols. The resulting antibody was biotinylated and finally dialyzed against PBS. Thus, this newly prepared anti-survivin antibody was suitable for application in spPLA.

### Development of solid-phase proximity ligation assay for survivin detection

For straightforward development of a proximity ligation assay (PLA) the TaqMan Protein Assay® offers an open kit variant whereby only a biotinylated antibody is required. This antibody must be sufficient for the use in PLA especially in terms of epitope recognition. Additionally, the target itself is also required for absolute quantification. First, the TaqMan Protein Assay® was used to achieve a homogenous PLA. Initially, problems with the assay performance were encountered when using urine samples directly, indicating a possible inhibition of the PCR reaction by matrix-based components. Since both heat or desalting treatment of urine had no positive effect on the assay performance, the cause of matrix-based interference appeared to be non-trivial. To eliminate these matrix-based interferences the TaqMan Protein Assay® was modified for application in spPLA according to published protocols [28,36]. For this, streptavidin-coated magnetic beads were used to attach a biotinylated portion of the previously purified anti-survivin antibodies. These antibody-functionalized magnetic beads served as a fishing rod for enrichment of survivin from urine. The detection of survivin by spPLA required only 500 μL of voided urine. This is a significant improvement compared to our survivin ELISA, which was based on urinary pellet cells from an average of 30 ml urine. Spike-in experiments showed an acceptable recovery rate between 68–78% for different survivin concentrations ranging from 15–1000 pg/mL. Urine samples without survivin addition showed measured values below the limit of detection. The use of tubes and tips with low DNA-binding affinity during all working steps led to an additional improvement of signal quality. Next, the best conditions for each step regarding reaction time and temperature were assessed. Incubation over night at +4˚C for the initial fishing step and three hours at room temperature for the binding reaction of the proximity probes appeared to be the best assay conditions. Longer incubation times did not

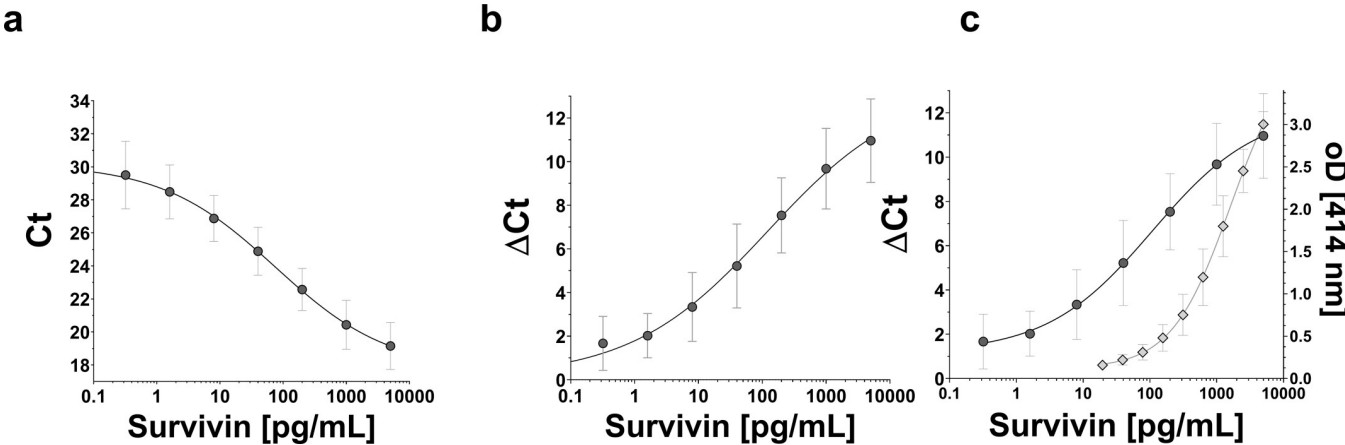

**Fig 1. Reference curve of the survivin spPLA based on 40 independent experiments.** Survivin concentrations ranged from 0.32 pg/mL to 5000 pg/mL. (**a**) Raw Ct-values versus survivin concentrations. (**b**) Delta Ct-values versus survivin concentrations. (**c**) Comparison of spPLA (solid) and ELISA (dotted, mean of 44 independent experiments) for survivin detection.

improve the signal quality significantly, whereas shorter incubation times showed a decrease in the signal-to-noise ratio. A mean reference curve of 40 independent experiments with corresponding standard deviation for survivin detection by the new developed spPLA based on the modified TaqMan Protein Assay® is shown in Fig 1.

The reference curve for survivin in Fig 1(A) shows a good signal-to-noise ratio with an average ΔCt of 11 comparing samples without survivin (Ct = 30) and with maximum survivin concentration of 5000 pg/mL (Ct = 19). Fig 1(B) depicts ΔCt values versus the survivin concentration of reference samples. The limit of detection was 1.45 pg/mL for survivin spPLA according to a four-parameter curve fitting using 40 independent blank measurements for the determination. The coefficient of variation for high, medium, and low survivin concentration was 7.45%, 5.19%, and 7.26%, respectively. In Fig 1(C) the new survivin spPLA is compared to the earlier published survivin ELISA. The limit of detection was substantially improved and decreased from 33 pg/mL to 1.45 pg/mL.

## Adjustment of the spPLA for survivin detection in urine

Ideally, the complete procedure for biomarker detection is carried out in a single reaction vessel. If multiple vessel-transferring steps are required, a potential risk for partial analyte loss must be kept in mind. Hence, the protocol was adapted to avoid as many transferring steps as possible, because the survivin concentration in real samples could be vanishing small. The final workflow is depicted in Fig 2.

To increase the chance of detecting a very small amount of survivin in voided urine, an initial concentration step was performed. Spike-in experiments showed that this step has no negative influence, like precipitation or accumulation, on the survivin protein indicating that this procedure is feasible. Subsequently, all following steps–neutralization, binding to magnetic beads, washing, binding of proximity probes and ligation of DNA strands–were performed in the same well of a microtiter plate. The final signal was detected by real-time PCR.

## Study population

The following results are based on urinary samples belonging to the same study group already been described by Gleichenhagen et al. [19]. A detailed description of the study group can be

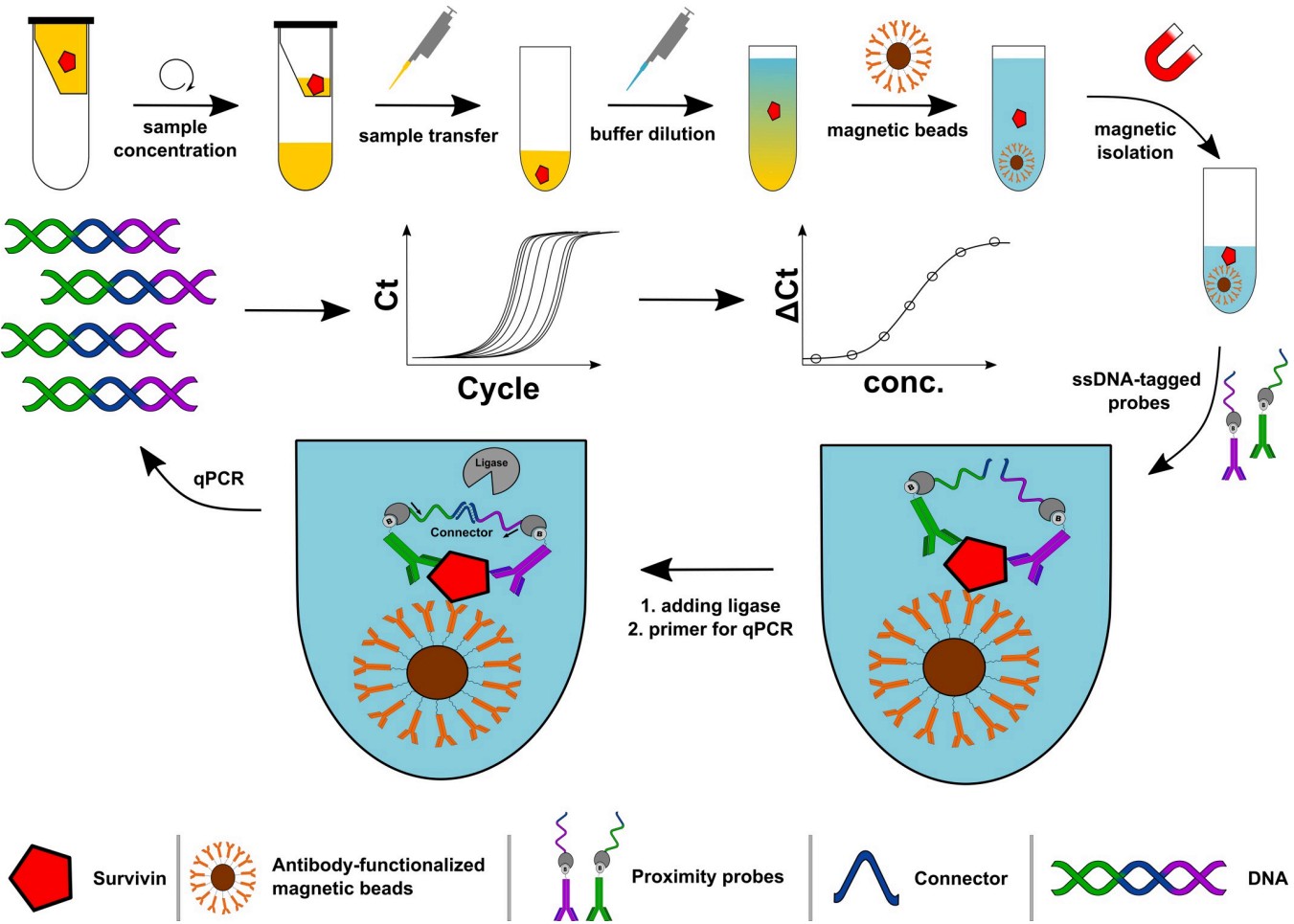

**Fig 2. Workflow for spPLA detection of survivin in urinary samples.** After urine concentration the remaining sample is diluted in buffer for pH neutralization. After sequential binding of survivin antibody-functionalized magnetic beads and proximity probes, DNA-strands come into close proximity. A connector allows the ligation, thereby forming a new chimeric DNA strand. Real-time PCR and further analysis allow protein quantification.

found in S1 and S2 Tables. The study included 243 participants, further divided into cases (n = 110) and clinical controls (n = 133). In contrast to previously published data, one case, a patient suffering from a low-grade tumor, was excluded due to exhausted sample material. Most participants were men (n = 174), the median age of the cases group and clinical control group was 74 and 71 years, respectively. The median volume of urine was 30 mL with a retention time of about 2 h within the bladder before voiding. Among all groups the median pH value of urine was 5. Additional information about the case group regarding tumor characteristics are listed in S2 Table.

## Assessment of survivin spPLA for detection of bladder cancer in urinary samples

The following results of the spPLA for survivin are based on voided urine samples. Previous spike-in experiments have suggested no negative effect on survivin stability from storage of voided urine at -20°C [19]. The corresponding urinary pellet was no longer available as all sample material was consumed during prior ELISA experiments, precluding a direct comparison of both methods [19]. The cutoff for ruling a sample positive was set to the limit of

**Table 1. Comparison of different cutoffs for the surviving spPLA.**

| Cutoff [pg/mL] | Sensitivity (%) | Specificity (%) | True-Positive(n) | True-Negative (n) | False-Positive (n) | False-Negative (n) |
|---|---|---|---|---|---|---|
| 1.45 | 30.0 | 89.5 | 33 | 119 | 14 | 77 |
| 2.55 | 24.5 | 94.8 | 27 | 126 | 7 | 83 |
| 4.15 | 20.0 | 97.8 | 22 | 131 | 2 | 88 |

detection (1.45 pg/mL), based on the fact that in non-malignant samples survivin should not be present or detectable [12]. In addition, Table 1 shows that a higher cutoff would unfavorably reduce the assay performance. The corresponding ROC curve is shown in S1 Fig.

In 33 case samples the spPLA detected survivin concentrations above the cutoff of 1.45 pg/mL, while 14 clinical controls had values above the cutoff. Accordingly, 81% of all detected survivin concentrations are below limit of detection and survivin is less frequently measurable in controls than in cases (p-value = 0.0001). Survivin spPLA shows a sensitivity of 30% and a specificity of 89% for the detection of bladder cancer in voided urine. The survivin concentrations of both groups are depicted in Fig 3.

A more differentiated group analysis with respect to tumor grading and tumor staging showed no difference in sensitivity. In contrast to the survivin ELISA the survivin spPLA was able to detect survivin in minimal volumes of voided urine as shown by S2 Fig. Logistic regression analyses showed no influence of microhematuria, retention time, or creatinine on survivin concentrations > 1.45 pg/mL. However, the chance for a detection of survivin in cases was approximately four times higher than in clinical controls (p-value < 0.001).

## Survivin as complementary marker for the detection of bladder cancer

Previously published work showed that survivin was not sufficient as a single biomarker for the detection of bladder cancer, but may be useful as a complementary marker as part of a marker panel [19]. For comparison, the results for survivin detection by spPLA were combined with results from the UBC® Rapid assay. The UBC® Rapid assay is a commercially available point-of-care test measuring fragments of cytokeratin 8 and 18 in voided urine. As shown by a Venn diagram (Fig 4), 16 cases were exclusively detected by the spPLA and 44 cases by UBC® Rapid, respectively. Both assays commonly detect only 17 cases. The combination of both assays yields a sensitivity of 70%, but showing an overall specificity of 86%, mainly due to false-positive results of the spPLA for survivin.

## Discussion

Bladder cancer still remains one of the most common cancers worldwide, with approximately 81% of cases due to known risk factors such as smoking or exposure to aromatic amines. Therefore, bladder cancer is a prime candidate for early detection and prevention strategies [1,2]. Currently, the gold standard for bladder cancer detection is cystoscopy, often complemented by cytology. Those methods are routinely applied if symptoms suspicious for bladder cancer are present like hematuria [37]. Typically, these symptoms are initially recognized by the patient himself, occur in later stages of tumor development and thereby lowering treatment options as well as chances for successful therapy. An advantage of biomarkers is their potential to detect cancer at much earlier stages while reducing invasive diagnostic procedures. Hence, biomarkers could facilitate an earlier and therefore more curative therapy, ideally resulting in a decreased mortality [38].

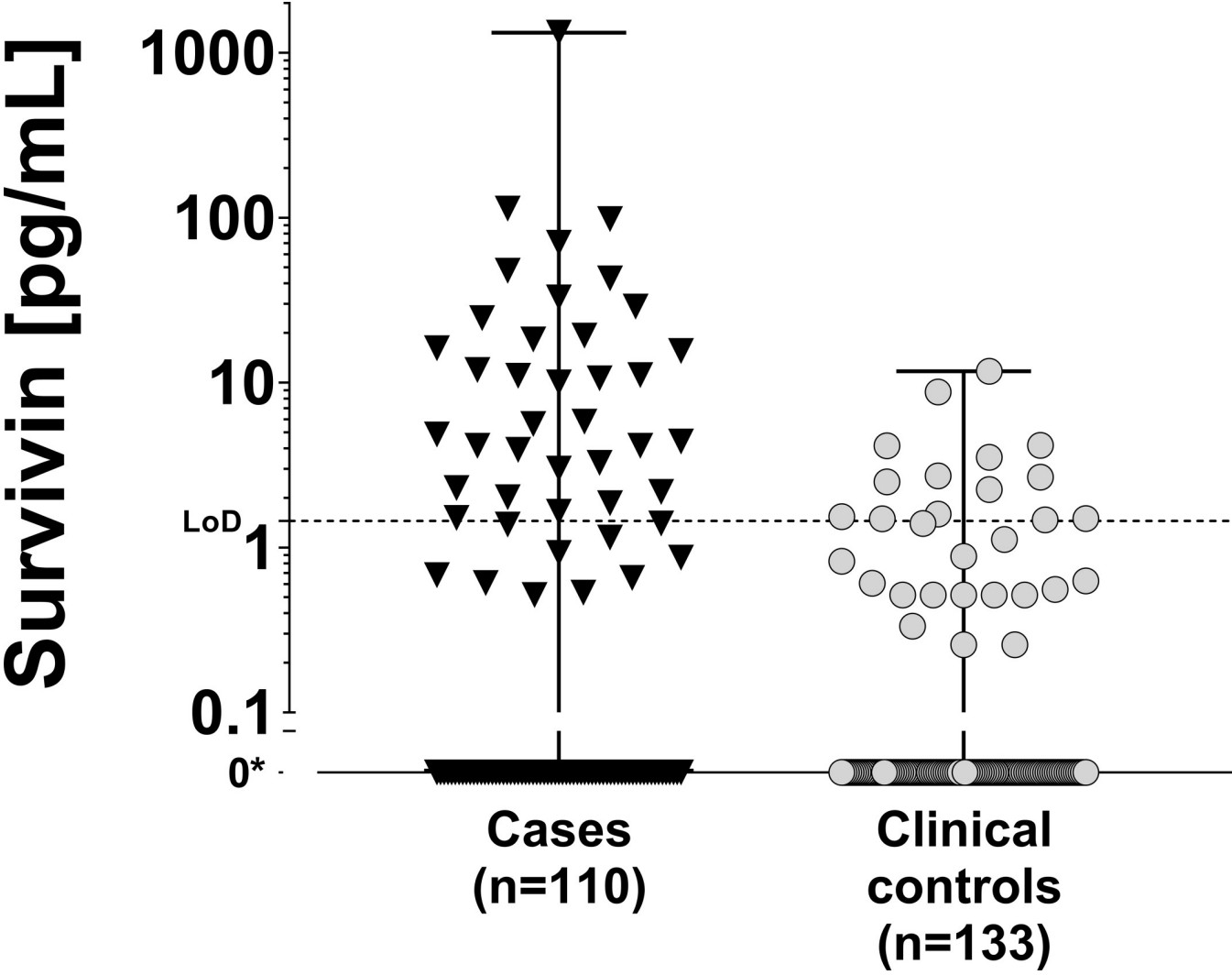

**Fig 3. Group analysis of spPLA survivin measurements.** Detectable amounts of survivin are depicted as dot plots on a logarithmic scale for cases and clinical controls with median and range. Samples containing no measurable amounts of survivin are indicated with 0*. LoD: Limit of detection.

Urine is a reasonable and ideal source for non-invasive biomarkers used to monitor or detect bladder cancer [4,8]. Due to the relatively high incidence of bladder cancer, it is of particular medical and economic interest to offer detection methods that are easy to apply, reliable and do not expose patients to unnecessary physical stress. For instance, German workers who were occupationally exposed to known bladder cancer carcinogens like aromatic amines are offered free medical examinations for bladder cancer screening [39–42].

In the past decades, the most frequently investigated urine-based biomarkers for the detection of bladder cancer have been UroVysion, uCyt+, and NMP22. Known drawbacks of these methods include observer bias, time-consuming procedures, confounders, and/or large variations in reported performance [40,43–45]. Within the last years the spectrum of commercially available marker tests has been extended to include Xpert® Bladder Cancer, CellDetect®, and UBC® Rapid. Those tests are currently under investigation regarding application into clinical routine [5,7,46]. The majority of new markers rely on mutation analysis or differential mRNA expression of certain genes. Although nucleic acid-based technologies represent a promising

# Cases

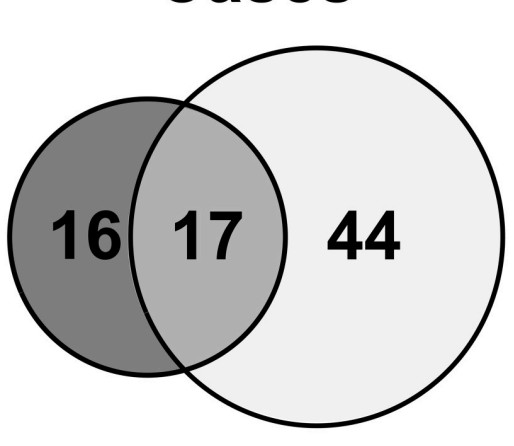

# Clinical controls

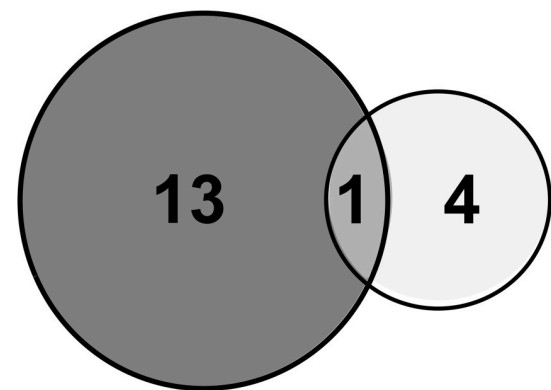

**Fig 4. Venn diagram for survivin spPLA and UBC Rapid for both groups, cases and clinical controls.** The cutoff for the spPLA was 1.45 pg/mL, resulting in 33 positive cases and 14 positive clinical controls. The cutoff for UBC Rapid assay was set to 10 mg/L, resulting in 61 positive cases and 5 positive clinical controls.

application for routine clinical diagnostics, profiling abnormally present proteins in human fluids or tissues is still one of the most widely used applications to assess a patient's health status. These applications are often straightforward and widely accepted as reliable detection methods because expressed proteins represent the functional endpoint of gene expression and are major players at the molecular level of a cell. The most sought-after targets in cancer biomarkers are differentially expressed in cancer versus normal cells and, indeed, survivin can be highly abundant in cancer, yet is absent from most normal somatic cells [12,47]. The ELISA is one of the most commonly used techniques to quantify proteins in human fluids. It is an easy to handle and affordable method, but sometimes has some drawbacks regarding the limit of detection. The original survivin ELISA developed by our group showed only a moderate sensitivity for bladder cancer detection [19]. A possible reason for this lack of sensitivity could have been of technical origin, i.e., limited ability to detect rare amounts of survivin.

A modern technique–the immuno-PCR–is able to transform a singular protein signal to an exponentially amplifiable DNA-based signal. This feature allows to detect vanishing small amounts of a target molecule [48]. Hence, the utility of immuno-PCR for survivin detection in urinary samples was explored using a commercially available and customizable proximity ligation assay.

The minimal requirements for quantification according to the TaqMan Protein Assay$^{®}$ is a $\Delta Ct \geq 3$ and was more than fulfilled with a $\Delta Ct$ of 11 by the newly developed assay. The spPLA for survivin resulted in a limit of detection of 1.45 pg/mL, which is about 23-times better than the limit of detection of the survivin ELISA [19]. This constitutes a major improvement for survivin detection and is in accordance with data from the literature for immuno-PCR assays showing an improved limit of detection for different protein targets [30,48,49]. Initial experiments of a homogenous PLA failed, most likely due to inhibiting elements in urine [6]. A prominent inhibitor for PCR is urea that causes denaturation of DNA polymerase, the key enzyme of the PCR and thus the primary signal generating step in the PLA reaction

[50,51]. The presence of metal ions originated from the urinary specimens could also had have a negative influence on the PLA. Unfortunately, desalting of the samples did not achieve an elimination of potentially inhibitors and we had to resort to an alternative approach based on magnetic beads. Switching to a solid phase-based assay provides the advantage of washing steps to remove all inhibiting elements present in urinary specimen prior to downstream applications. In addition, spike-in experiments confirmed that the assay was applicable to detect and quantify survivin in urinary samples and, furthermore, is not prone to non-specific interactions with other urine components. A major advantage over our previous survivin ELISA is that the spPLA can detect minimal amounts of survivin in only 500 μL of voided urine rather than in urine cell pellets. In contrast, the survivin ELISA was nearly unable to detect survivin in voided urine of bladder cancer patients (4/111). Many other analytical assays also depend on urine cell pellets as the analyte source. The most plausible reason for this is that tumor cells shed into the urine are enriched. Therefore, the urine cell pellet provides perhaps the best source for the detection of a biomarker. Consequently, the urine pellet is a highly sought-after material for any type of analysis, whether as part of a study or in routine clinical practice. However, it is desirable to measure multiple biomarkers in a single sample to improve the performance of a panel. Using pelleted urine cells, the sample material is very limited. Thus, we aimed to develop a highly sensitive method to detect survivin in voided urine. The spPLA method for survivin is a method that could overcome practical limitations such as small sample volumes, which at the same time allows survivin to be more easily measured in parallel with other methods/markers in future research projects.

To test the usefulness of spPLA for survivin under realistic conditions, survivin concentration was measured in urine samples from bladder cancer patients and clinical controls. The survivin spPLA was able to detect 30% of all bladder cancer cases accompanying 14 false-positive test results yielding a specificity of 89%. Despite that the limit of detection for survivin spPLA was markedly improved, the survivin concentration in 77 bladder cancer samples could still not be detected. Possibly the technical hurdles for survivin detection in urine are even more challenging than originally assumed. The concentration of survivin molecules shed into urine could be too low for reliable binding to our survivin antibody or the binding might have been inhibited by (residual) components of the matrix, e.g. residual amounts of urea. To counteract this weakness, the capture antibody coupled to the magnetic beads could be replaced with an alternative antibody for survivin from different sources in future studies. In addition, one could also consider changing the molecular nature of the binders for survivin. Nowadays, a new kind of target-specific binders become more accessible for biomarker applications. These binders are DNA- or RNA-based aptamers facilitating completely new opportunities for assay development [52–55]. Unlike antibodies, aptamers can be generated/selected under user-defined parameters. This allows a much broader range of possible buffer compositions, especially in terms of chemical components and pH [53].

The specificity of the survivin spPLA was lower (89.5%) than the specificity observed using the survivin ELISA (98%) [19]. Inflammation could be a possible confounder and might explain the presence of survivin in cancer-free patients [56]. A methodological explanation for the reduced specificity could be false-positive signal amplification due to the PCR reaction of spPLA. This is a possible drawback of immuno-PCR techniques [28,32,33]. To minimize the risk for such effects, the survivin antibody was antigen-purified and moreover antibody-functionalized beads allowed several washing steps to even more lowering the risk for false-positive signals. Hence, the reason for the reduced specificity is unknown and remains to be elucidated in future work.

In accordance with previous results survivin again showed its potential as a complementary marker in combination with UBC® Rapid by increasing the sensitivity for bladder cancer

detection to 70% at a specificity of 86%. The exclusively detected 16 bladder cancer cases by survivin spPLA underlines the benefit of this marker. The validation of biomarkers for early detection of cancer requires a prospective study design that provides serial samples from participants of a high-risk cohort that are initially asymptomatic. The obtained pre-diagnostic samples should represent earlier stages of tumors, which are the basis to test marker candidates. Previous studies on other cancer entities have indicated that the time window for early detection is not much more than one year [57,58]. That interval might be different for bladder cancer, but some prospective studies suggest a similar window for early detection in bladder cancer [40,59]. Currently, survivin is part of the prospective UroFollow study [46]. Upcoming results may contribute to a better understanding of survivin and its role during tumor formation and as an early marker.

Limitations of the presented study are the case-control design with a relatively small sample size and the selection of the control group. The current study was intended as an initial assessment to explore cutting edge technology for the detection of biomarkers of low abundance in human fluids, especially urine. Another limitation of this study is that the original urine cell pellet was completely consumed by the previous ELISA analyses. Therefore, a direct comparison between both methods based on the same biomarker source was unfortunately not possible.

## Conclusion

In summary, the utility of survivin detection by spPLA technique in urine was explored. The underlying hypothesis that diagnostic sensitivity is due to technical limitations of survivin detection could not be confirmed. Even conversion of a singular protein signal into an amplifiable DNA-based signal failed to improve the performance of bladder cancer detection by survivin alone. However, by assessing survivin spPLA, we demonstrated that this method is suitable for the detection of low protein levels in minimal urine sample volumes. Furthermore, the combination of UBC® Rapid and survivin spPLA again demonstrated the value of survivin as a complementary marker for the detection of bladder cancer. Further research investigations could focus on multiplex assays including survivin as well as other candidate protein biomarkers, as this modified immuno-PCR method may also be applied to other protein biomarkers. In addition, the design of the herein applied spPLA for survivin allows future applications using liquid biopsies of various origins to further explore the utility of survivin as biomarker.

## Supporting information

**S1 Fig. ROC curve of the survivin spPLA assay.**
(TIF)

**S2 Fig. Dot blots of survivin concentrations.** Detection of survivin in 500 μL voided urine from bladder cancer patients measuered by (**a**) spPLA and (**b**) ELISA. Samples containing no measurable amounts of survivin are indicated with 0*. LoD: Limit of Detection.
(TIF)

**S1 Table. Characteristics of the study group.**
(DOCX)

**S2 Table. Characteristics of the 110 bladder cancer cases.**
(DOCX)

## Acknowledgments

We thank Beate Pesch and Daniel G. Weber for constructive discussion on study and assay development.

## Author Contributions

**Conceptualization:** Thorsten Ecke, Georg Johnen.

**Data curation:** Jan Gleichenhagen, Christian Arndt.

**Formal analysis:** Swaantje Casjens.

**Investigation:** Jan Gleichenhagen, Christian Arndt, Carmen Töpfer, Irina Raiko.

**Methodology:** Jan Gleichenhagen, Christian Arndt, Georg Johnen.

**Project administration:** Holger Gerullis, Thorsten Ecke, Georg Johnen.

**Supervision:** Holger Gerullis, Dirk Taeger, Thorsten Ecke, Thomas Brüning, Georg Johnen.

**Visualization:** Jan Gleichenhagen.

**Writing – original draft:** Jan Gleichenhagen, Swaantje Casjens.

**Writing – review & editing:** Jan Gleichenhagen, Irina Raiko, Dirk Taeger, Thorsten Ecke, Thomas Brüning, Georg Johnen.

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
