## [Decision Letter · Decision Letter 0]

11 Apr 2022

PONE-D-22-04954Exploring solid-phase proximity ligation assay for survivin detection in urinePLOS ONE

Dear Dr. Gleichenhagen,

Thank you for submitting your manuscript to PLOS ONE. After careful consideration, we feel that it has merit but does not fully meet PLOS ONE’s publication criteria as it currently stands. Therefore, we invite you to submit a revised version of the manuscript that addresses the points raised during the review process.

We look forward to receiving your revised manuscript.

Kind regards,

Katerina Kourentzi, PhD

Academic Editor

PLOS ONE

Journal Requirements:

2. Please note that in order to use the direct billing option the corresponding author must be affiliated with the chosen institute. Please either amend your manuscript to change the affiliation or corresponding author, or email us at plosone@plos.org with a request to remove this option.

Reviewers' comments:

Reviewer's Responses to Questions

**Comments to the Author**

1. Is the manuscript technically sound, and do the data support the conclusions?

Reviewer #1: Partly

Reviewer #2: Yes

2. Has the statistical analysis been performed appropriately and rigorously? 

Reviewer #1: Yes

Reviewer #2: Yes

3. Have the authors made all data underlying the findings in their manuscript fully available?

Reviewer #1: Yes

Reviewer #2: Yes

4. Is the manuscript presented in an intelligible fashion and written in standard English?

Reviewer #1: Yes

Reviewer #2: Yes

5. Review Comments to the Author

Reviewer #1: The author used the principal of proximity ligation assay to detect survivin in urine by a method named solid-phase proximity ligation assay (spPLA). The manuscript is mainly about a method development and more likely would fit in a method-based journal. However, there are some concerns that need to be addressed in the manuscript.

Major concern:

1. Both the sensitivity and specificity of the UBC rapid test itself for Survivin detection is better than the developed SpPLA method developed by the authors. Although combining both the methods increases the sensitivity, the specificity is overall decreased (Fig. 4). Therefore, the feasibility of the spPLA test in addition to UBC test for survivin detection is a concern.

Minor concern:

2. As the author compared the UBC rapid assay and spPLA in clinical samples, they need to explain the UBC rapid assay in brief somewhere in the text.

3. Does the CT/deltaCT valeu detected by spPLA correlates with the different stage of bladder cancer?

Reviewer #2: The authors in this paper adapt solid-phase proximity ligation assay (spPLA) for survivin detection in urine. The paper The authors stated that the motivation of this approach was to overcome the sensitivity limitation of their in-house ELISA. However, their new survivin spPLA showed a sensitivity of 30% and specificity of 89%, which is worse than their survivin ELISA’s sensitivity of 35% and specificity of 98%, published previously. In combination with UBC Rapid, the spPLA (sensitivity of 70%, specificity of 86%) did not show a clear improvement over ELISA and UBC Rapid combination (sensitivity of 66%, specificity of 95%). Figure 1c showed that spPLA has significantly better analytical sensitivity than ELISA; however, spPLA has lower clinical sensitivity. The authors mentioned that the urine samples were stored at 20C until analyzed; this raises the question of whether freezing affected urinary proteins. spPLA also has higher variability. In the workflow of spPLA, after concentrating with a spin column, the sample was diluted ~10X with PBS-BSA before magnetic bead capture. This neutralization step might reduce the sensitivity significantly. A more concentrated neutralization buffer with less volume would minimize the dilution effect. spPLA for survivin yields similar performance as ELISA. However, spPLA is more complex, time-consuming, and requires specialized reagents and instrument, minimizing the benefits of spPLA for bladder cancer detection. It is not clear whether the low clinical sensitivity of spPLA for bladder cancer diagnostic was due to the limitation of spPLA or the limitation of survivin as a lone biomarker for bladder cancer. A multiplex assay detecting survivin and other known biomarkers for bladder cancer may improve the clinical performance.

In summary, this work is well done, the paper is well written with clear experimental steps, and the results are of good quality and very well explained. However, the potential utility of this work is not compelling.

6. PLOS authors have the option to publish the peer review history of their article (what does this mean?). If published, this will include your full peer review and any attached files.

Reviewer #1: No

Reviewer #2: **Yes: **Binh Vu

---

## [Author Response · Author response to Decision Letter 0]

24 May 2022

Dear Editor,

thank you for considering our manuscript for publication in PLOS One. 

We thank both reviewers for their valued comments. According to the reviewer suggestions we revised our manuscript in order to emphasize the methodical focus of our work presenting a useful and very sensitive tool for the detection of survivin in urine, even when small sample amounts are available.

Please find below our point-to-point response. Corresponding changes are indicated by line number. 

Reviewer #1: 

The author used the principal of proximity ligation assay to detect survivin in urine by a method named solid-phase proximity ligation assay (spPLA). The manuscript is mainly about a method development and more likely would fit in a method-based journal. However, there are some concerns that need to be addressed in the manuscript.

Major concern:

1. Both the sensitivity and specificity of the UBC rapid test itself for Survivin detection is better than the developed SpPLA method developed by the authors. 

- UBC Rapid detects the biomarkers KRT 8 and KRT 18 (but not survivin), whereas the biomarker survivin is detected by the newly developed spPLA. This is stated more clearly in line 347.

Although combining both the methods increases the sensitivity, the specificity is overall decreased (Fig. 4). Therefore, the feasibility of the spPLA test in addition to UBC test for survivin detection is a concern.

- We agree with the reviewer that spPLA for survivin in its current form does not improve the overall diagnostic performance. However, a limiting factor can be the volume of voided urine and thus the number of cells that can be obtained. A sufficient number of cells is necessary for cytology, FISH (UroVysion etc.), and other tests, and was also required for previous survivin assays (e.g., based on the measurement of survivin protein or the less stable mRNA). We therefore aimed at developing a method that would require less sample material than the classical ELISA we investigated previously (added to discussion at line 418f). Our approach shows that it is possible to reduce the required sample volume and additionally improve the analytical detection limit, making spPLA to a promising method. To the best of our knowledge, we describe for the first time the application of an spPLA method using urine for the detection of a protein biomarker. However, biomarker performance is not solely dependent on methodological parameters, but also on tumor biology. The fact that survivin failed to improve clinical diagnostics despite a lower detection limit suggests that voided urine may show too low concentrations of survivin, based on unknown biological factors. However, we showed that an immunoPCR-based method can be used for protein detection in minimal urine samples as well as improve the detection limit. To state this point more clearly, we rephrased our conclusion.

Minor concern:

2. As the author compared the UBC rapid assay and spPLA in clinical samples, they need to explain the UBC rapid assay in brief somewhere in the text.

- We added a passage explaining UBC Rapid (line 347) 

3. Does the CT/deltaCT valeu detected by spPLA correlates with the different stage of bladder cancer?

- No correlation was observed. This information was added to the manuscript in line 336.

Reviewer #2: 

The authors in this paper adapt solid-phase proximity ligation assay (spPLA) for survivin detection in urine. The paper The authors stated that the motivation of this approach was to overcome the sensitivity limitation of their in-house ELISA. 

However, their new survivin spPLA showed a sensitivity of 30% and specificity of 89%, which is worse than their survivin ELISA’s sensitivity of 35% and specificity of 98%, published previously. In combination with UBC Rapid, the spPLA (sensitivity of 70%, specificity of 86%) did not show a clear improvement over ELISA and UBC Rapid combination (sensitivity of 66%, specificity of 95%). Figure 1c showed that spPLA has significantly better analytical sensitivity than ELISA; however, spPLA has lower clinical sensitivity. 

- At this point it should be clarified that the results of ELISA and immunoPCR are based on different matrices. Survivin detection by ELISA is based on a harvested cell pellet from urine, which requires a much larger sample volume (usually about 30 ml) than spPLA, which is measured in only 0.5 ml of voided urine. Because the volume and thus the cell pellet is a limiting factor (especially when it is also used for other diagnostic methods like cytology or FISH), we tried to develop a new method that would require a smaller volume of urine. With the classical ELISA, no survivin could be detected in the same volume of voided urine we used with the new method. Thus, the advantage of spPLA is that almost the same performance as ELISA can be achieved with drastically less sample material (see results line 317ff and discussion line 411ff). Therefor we added an additional supplemental figure (S2).

The authors mentioned that the urine samples were stored at 20C until analyzed; this raises the question of whether freezing affected urinary proteins. 

- Freezing the samples can of course have a negative effect on the proteins. Due to its natural function as a "shuttle protein" with many interaction partners, it is of course conceivable that survivin accumulates with these and may no longer be free or detectable in the sample. However, we did not observe such effects on survivin. In our previous study, we investigated this effect in spike-in experiments. No significant effect on survivin could be detected after samples were stored at -20°C and thawed again. In addition, it should be considered that recombinant survivin, previously isolated in a denaturing manner and subsequently renatured, is stable at -20°C for a very long time. This is another indication that survivin should be detectable in samples stored at -20°C. Added information at line 315ff.

spPLA also has higher variability. In the workflow of spPLA, after concentrating with a spin column, the sample was diluted ~10X with PBS-BSA before magnetic bead capture. This neutralization step might reduce the sensitivity significantly. A more concentrated neutralization buffer with less volume would minimize the dilution effect.

- The idea of a concentrated neutralization buffer with a smaller volume is very interesting and will be considered in our future work. We appreciate this idea.

spPLA for survivin yields similar performance as ELISA. However, spPLA is more complex, time-consuming, and requires specialized reagents and instrument, minimizing the benefits of spPLA for bladder cancer detection. It is not clear whether the low clinical sensitivity of spPLA for bladder cancer diagnostic was due to the limitation of spPLA or the limitation of survivin as a lone biomarker for bladder cancer. A multiplex assay detecting survivin and other known biomarkers for bladder cancer may improve the clinical performance.

- We agree that a multiplex-based approach would definitely be better for clinical diagnostics. Multiplexing and marker combinations were also a reason to develop an assay that uses less sample material and therefore leaving enough material for other assays. The current workflow, as the reviewer correctly pointed out, is too complex and not suitable for routine diagnostics. This requires further optimization of the individual steps, automation and possibly also direct combination with other markers in a multiplex assay. Here we established the methodological basis for research options of course seeking for a multiplex assay. Survivin again showed its potential as a complementary marker therefore seems to be ideal for implementation in a multiplex-based assay. This point has been added to the discussion at line 474ff.

In summary, this work is well done, the paper is well written with clear experimental steps, and the results are of good quality and very well explained. However, the potential utility of this work is not compelling.

- The potential benefit of this work is that the developed spPLA method is feasible to detect proteins in complex matrices, like urine. The advantages of the method are the required small sample volumes and the low detection limit.

---

## [Decision Letter · Decision Letter 1]

12 Jun 2022

Exploring solid-phase proximity ligation assay for survivin detection in urine

PONE-D-22-04954R1

Dear Dr. Gleichenhagen,

We’re pleased to inform you that your manuscript has been judged scientifically suitable for publication and will be formally accepted for publication once it meets all outstanding technical requirements.

Kind regards,

Katerina Kourentzi, PhD

Academic Editor

PLOS ONE

Additional Editor Comments (optional):

Reviewers' comments:

Reviewer's Responses to Questions

**Comments to the Author**

1. If the authors have adequately addressed your comments raised in a previous round of review and you feel that this manuscript is now acceptable for publication, you may indicate that here to bypass the “Comments to the Author” section, enter your conflict of interest statement in the “Confidential to Editor” section, and submit your "Accept" recommendation.

Reviewer #1: All comments have been addressed

Reviewer #2: All comments have been addressed

2. Is the manuscript technically sound, and do the data support the conclusions?

Reviewer #1: Yes

Reviewer #2: Yes

3. Has the statistical analysis been performed appropriately and rigorously? 

Reviewer #1: Yes

Reviewer #2: Yes

4. Have the authors made all data underlying the findings in their manuscript fully available?

Reviewer #1: Yes

Reviewer #2: Yes

5. Is the manuscript presented in an intelligible fashion and written in standard English?

Reviewer #1: Yes

Reviewer #2: Yes

6. Review Comments to the Author

Reviewer #1: The manuscript used an assay to detect survivin a urinary protein associated with bladder cancer. The reproducibility of the assay is not clear. Although the method in its current advancement is not optimized enough for clinical detection of bladder cancer, further improvement needs to be done. However, publication of the manuscript will help in understanding and gaining interest in the assay.

Reviewer #2: All concerns were addressed and the explanations were clear. The materials and methods in this paper would be useful for developing other biomarkers.

7. PLOS authors have the option to publish the peer review history of their article (what does this mean?). If published, this will include your full peer review and any attached files.

Reviewer #1: No

Reviewer #2: No

---

## [Editor Report · Acceptance letter]

21 Jun 2022

PONE-D-22-04954R1 

Exploring solid-phase proximity ligation assay for survivin detection in urine 

Dear Dr. Gleichenhagen:

I'm pleased to inform you that your manuscript has been deemed suitable for publication in PLOS ONE. Congratulations! Your manuscript is now with our production department. 

Kind regards, 

on behalf of

Dr. Katerina Kourentzi 

Academic Editor

PLOS ONE